

# Exact gravity duals for simple quantum circuits

Johanna Erdmenger[1], Mario Flory[2], Marius Gerbershagen[1⋆],
Michal P. Heller[3,4] and Anna-Lena Weigel[1]

**1** Institute for Theoretical Physics and Astrophysics and Würzburg-Dresden Cluster of
Excellence ct.qmat, Julius-Maximilians-Universität Würzburg,
97074 Würzburg, Germany
**2** Instituto de Física Teórica UAM-CSIC, c/ Nicolás Cabrera 13-15, 28049, Madrid, Spain
**3** Max Planck Institute for Gravitational Physics (Albert Einstein Institute),
14476 Potsdam-Golm, Germany
**4** Department of Physics and Astronomy, Ghent University, 9000 Ghent, Belgium

⋆ marius.gerbershagen@physik.uni-wuerzburg.de

## Abstract

Holographic complexity proposals have sparked interest in quantifying the cost of state preparation in quantum field theories and its possible dual gravitational manifestations. The most basic ingredient in defining complexity is the notion of a class of circuits that, when acting on a given reference state, all produce a desired target state. In the present work we build on studies of circuits performing local conformal transformations in general two-dimensional conformal field theories and construct the exact gravity dual to such circuits. In our approach to holographic complexity, the gravity dual to the optimal circuit is the one that minimizes an externally chosen cost assigned to each circuit. Our results provide a basis for studying exact gravity duals to circuit costs from first principles.

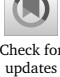

# 1 Introduction

The last couple of years have witnessed a substantial progress in the study of complexity measures of quantum circuits both in quantum field theory and in holographic bulk prescriptions (see [1] for a review). However, these two approaches have largely remained separate, with only conjectural or qualitative connections between the two sides established.

Our paper aims to identify a scenario where such a connection can be made in a robust manner, by constructing an explicit gravity dual to a simple quantum circuit in holographic quantum field theories. We will achieve this by focusing on local conformal transformations in the setting of the $AdS_3/CFT_2$ correspondence [2–4]. This has already proven to be a fruitful ground for complexity research on the gravity [5–8] and the field theory [9–13] sides of the correspondence. Our goal is to bridge the two perspectives by constructing an explicit gravity dual to a sequence of local conformal transformations acting on the vacuum state.

On the bulk side, the solutions associated to local conformal transformations of the vacuum are the Bañados geometries [14]. On the boundary side, following [9], we will consider circuits originating from the action of the exponentiated holomorphic component of the energy-momentum tensor (or equivalently the antiholomorphic one). These circuits will be taken to act on the vacuum state. The setup can be thought of as performing a local conformal transformation in a gradual fashion. In [9] and the subsequent works [10–13], significant insights were gained into quantifying the complexity of such a process. Due to the use of conformal symmetry, these circuits are particularly well-suited for a holographic mapping to gravity.

We perform operations in a gradual way, i.e. as a sequence of operations indexed by a circuit parameter that determines where we are in this process. In the holographic complexity literature, one typically thinks about the circuit parameter as an auxiliary variable. The key idea of our work is to identify this parameter with the physical time on the asymptotic boundary and to use the boundary geometry to trigger the gradual state preparation of interest. While we are not the first to advocate the use of physical time as a circuit parameter (see, in particular, [5, 15]), the novel aspect of our work is the full control we gain over both the circuit and the dual geometry.

# 2 A simple quantum circuit

In this section, we describe the construction of circuits implementing conformal transformations in a gradual way. We will construct two circuits implementing this idea distinguished by the interpretation of the circuit time parameter. In the first construction (case (a)), the circuit parameter $\tau$ is an auxiliary parameter independent on the physical time coordinate $t$ on the manifold in which the conformal field theory lives, while in the second construction (case (b)) the physical time coordinate and circuit parameter are identical.

We work in a two-dimensional conformal field theory in Euclidean signature on a unit-radius spatial circle parametrized by $\phi$. In both constructions, the circuit starts with a reference

state, which we take to be the vacuum $|0\rangle$ and then changes the state by acting on it with an operator formed from Virasoro algebra generators $L_n$,

$$|\psi(\tau)\rangle = U(\tau)|0\rangle, \quad \text{with} \quad U(\tau) = \mathcal{P}\exp\left(-\int_0^\tau d\tilde{\tau}\, Q(\tilde{\tau})\right), \tag{1}$$

and

$$Q(\tau) = \sum_n \epsilon_{-n}(\tau)L_n. \tag{2}$$

$U(\tau)$ is the analytic continuation of a unitary operator to Euclidean signature. For simplicity, we consider only one copy of the Virasoro algebra. The circuit generator $Q(\tau)$ can be equivalently written by smearing the holomorphic component of the energy-momentum tensor,

$$Q(\tau) = \int_0^{2\pi} \frac{d\phi}{2\pi}\, T(z)\,\epsilon(\tau,z), \tag{3}$$

where $\epsilon(\tau,z) = \sum_n \epsilon_n(\tau)e^{nz}$ and $z = t + i\phi$ with $t$ being the Euclidean time variable. Throughout this publication we use the notation

$$T(z) = \sum_n L_n e^{nz}, \quad \bar{T}(\bar{z}) = \sum_n \bar{L}_n e^{n\bar{z}}, \tag{4}$$

where $\bar{L}_n$ are the generators of the second copy of the Virasoro algebra.

The circuit implements at each $\tau$ a conformal transformation $z \to f(\tau,z)$. The two constructions of the circuit mentioned above are distinguished by the value of $\epsilon(\tau,z)$. Since the conformal transformations form a group with group action realized by composition of functions, the parameter $\epsilon(\tau,z)$ is related to $f(\tau,z)$ by [9]

$$\epsilon(\tau,f(\tau,z)) = \frac{d}{d\tau}f(\tau,z). \tag{5}$$

If the circuit parameter $\tau$ is an auxiliary parameter independent of $z$ (case (a)), the solution of this equation is given by

$$\epsilon_{(a)}(\tau,z) = \dot{f}(\tau,F(\tau,z)), \tag{6}$$

where $F(\tau,z)$ is the inverse of $f(\tau,z)$ defined by $f(\tau,F(\tau,z)) = z$ and $\dot{f}$ denotes the derivative of $f$ w.r.t. its first argument, i.e. the $\tau$-derivative at a fixed value of $z$. On the other hand, if the circuit parameter $\tau$ is given by the physical time $t$ then the holomorphic coordinate $z$ that is transformed by the conformal transformations depends on $\tau$ such that the solution of (5) is given by

$$\epsilon_{(b)}(t,z) = \dot{f}(t,F(t,z)) + f'(t,F(t,z)), \tag{7}$$

where $f'$ denotes the derivative of $f$ w.r.t. its second argument. The energy-momentum tensor at circuit time $\tau$ in both constructions is given by

$$U^\dagger(\tau)T(z)U(\tau) = f'(\tau,z)^2 T(f(\tau,z)) + \frac{c}{12}\{f(\tau,z),z\}. \tag{8}$$

Therefore, the action of one layer of the circuit between some $\tau$ and $\tau + d\tau$ is as follows. If $\tau$ is treated as an independent auxiliary parameter, we get

$$e^{Q_{(a)}(\tau)d\tau}U^\dagger(\tau)T(z)U(\tau)e^{-Q_{(a)}(\tau)d\tau} = f'(\tau+d\tau,z)^2 T(f(\tau+d\tau,z)) + \frac{c}{12}\{f(\tau+d\tau,z),z\}, \tag{9}$$

while for $\tau = t$,

$$e^{Q_{(b)}(t)dt}U^\dagger(t)T(z)U(t)e^{-Q_{(b)}(t)dt} = f'(t+dt,z+dt)^2 T(f(t+dt,z+dt)) \tag{10}$$

$$+ \frac{c}{12}\{f(t+dt,z+dt),z\}. \tag{11}$$

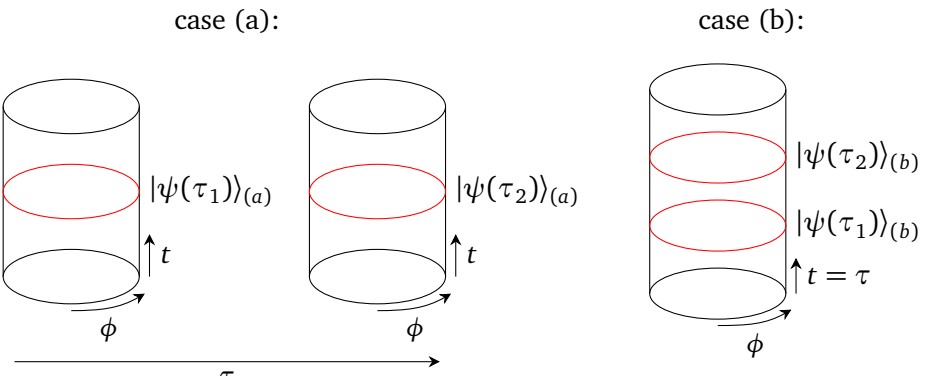

Figure 1: Depiction of the two circuits we consider. In case (a), the circuit evolution proceeds through a sequence of states living on time slices of different spacetimes (marked in red). There is no associated evolution with respect to physical time $t$. In case (b), the states live on different time slices of the same spacetime. In holography, in case (a) we have a sequence of independent gravity dual geometries, whereas in case (b) we arrive at a single gravity dual geometry.

Equations (9) and (11) further illustrate the difference between the two circuit constructions. In case (a) the state $|\psi(\tau)\rangle$ lives on the same time slice in physical time (say at $t = 0$) for all $\tau$. The circuit evolution in this case creates a sequence of states dual to Bañados geometries. On the other hand, in case (b) the states $|\psi(t)\rangle$ live on different time slices of the same geometry (see Fig. 1). Therefore, in this case time evolution also has to include evolution in the holomorphic coordinate $z$. Equations (9) and (11) may also be used to derive $Q_{(a)}$ and $Q_{(b)}$ by expanding to linear order in, respectively, $d\tau$ and $dt$ and applying the Virasoro algebra. This recovers, respectively, (6) and (7).

What we have described so far as case (a) is the setup considered in [9], while case (b) is the natural generalization to consider when implementing gradual conformal transformations in a single spacetime. Of course, the sequence of states described by these circuits differs between the two constructions since the circuit generators $Q_{(a)}$ and $Q_{(b)}$ are different. We will explore the consequences of these differences in the following section.

Before we close this section, let us expand on why it is particularly insightful to consider conformal transformations in the context of holographic complexity. To this end, it is important to emphasize that the energy-momentum tensor sector, up to the value of the central charge, is universal across all conformal field theories. Therefore, the current setup applies equally well to the Ising model conformal field theory and to holographic theories. Had we focused on, for example, circuits generated by a scalar operator, it would lead to a less universal setup. Furthermore, because of earlier efforts in [9–13], we know rather well what are the interesting possibilities for assigning a cost to (1-2) and, in several cases, what are the optimal circuits. Finally, the energy-momentum tensor couples to the metric in which a conformal field theory in question lives. This opens a possibility of triggering the circuit by placing the conformal field theory in an appropriately chosen geometry, which we discuss in the next section. Finally, in the case of gravity in three dimensional anti-de Sitter space, the boundary metric and the expectation value of the energy-momentum tensor allow to straightforwardly obtain the full bulk geometry. This is what we discuss in section 4.

## 3 Generating the same sequence of states using sources

### 3.1 General discussion

In general, physical implementations of quantum operations are based on time evolution of quantum systems. For the circuit from section 2, the operation is generated by the generator $Q(\tau)$ from (2). Our idea is to use the physical Hamiltonian of a conformal field theory living in some background metric $g_{ij}^{(0)}$,

$$H(t) = \int_0^{2\pi} \frac{d\phi}{2\pi} \sqrt{g^{(0)}} \, T^t{}_t \,, \tag{12}$$

see e.g. [16], to generate the circuit. Therefore, we demand

$$H(t) \overset{!}{=} Q(t) \,. \tag{13}$$

This identification allows us to derive the correct background metric $g_{ij}^{(0)}$ which triggers the conformal transformations applied at each time step of the circuit[1]. Therefore, this construction yields a single bulk geometry for the entire circuit. Because the Hamiltonian (12) is the generator of time translations in the physical time $t$, this construction is the natural method for deriving a bulk dual to the circuit (b) constructed in the previous section in which the circuit parameter $\tau$ is identified with $t$.

However, this method nevertheless allows us to write down a bulk dual to the circuit (a) consisting of a sequence of states living on different time slices of the same spacetime manifold. The two constructions in the implementation of these circuits derived in this section then differ only in the source configuration $g_{ij}^{(0)}$ as we will see below.

For the particular circuits from section 2, it turns out to be sufficient to choose a *flat* boundary metric as source, however, the choice of coordinate system becomes important. General flat metrics are parametrized by diffeomorphisms $(w(z,\bar{z}), \bar{w}(z,\bar{z}))$ dependent on both $z$ and $\bar{z}$,

$$ds_{(0)}^2 = dw d\bar{w} = \frac{\partial w}{\partial z}\frac{\partial \bar{w}}{\partial z} dz^2 + \left( \frac{\partial w}{\partial z}\frac{\partial \bar{w}}{\partial \bar{z}} + \frac{\partial w}{\partial \bar{z}}\frac{\partial \bar{w}}{\partial z} \right) dz d\bar{z} + \frac{\partial w}{\partial \bar{z}}\frac{\partial \bar{w}}{\partial \bar{z}} d\bar{z}^2 \,. \tag{14}$$

Our conventions follow these in the previous section and entail

$$z = t + i\phi \,, \quad \bar{z} = t - i\phi \,. \tag{15}$$

The constant time slices with respect to which the Hamiltonian $H(t)$ generates time evolution are defined by $z + \bar{z} = \text{const}$. Via (13), these are also lines of constant values of the circuit parameter.

Based on this definition for our metric, we now derive expressions for the diffeomorphisms $w(z,\bar{z})$ and $\bar{w}(z,\bar{z})$ in terms of the conformal transformations $f(t,z)$. For this purpose we express the Hamiltonian $H(t)$ in terms of Virasoro generators and demand equality with the circuit generator $Q(t)$, implementing (13). We apply the standard tensor transformation rules to obtain,

$$T_{zz} = T(w(z,\bar{z})) \left( \frac{\partial w}{\partial z} \right)^2 + \bar{T}(\bar{w}(z,\bar{z})) \left( \frac{\partial \bar{w}}{\partial z} \right)^2 \,,$$

$$T_{\bar{z}\bar{z}} = T(w(z,\bar{z})) \left( \frac{\partial w}{\partial \bar{z}} \right)^2 + \bar{T}(\bar{w}(z,\bar{z})) \left( \frac{\partial \bar{w}}{\partial \bar{z}} \right)^2 \,, \tag{16}$$

$$T_{z\bar{z}} = T(w(z,\bar{z})) \frac{\partial w}{\partial z}\frac{\partial w}{\partial \bar{z}} + \bar{T}(\bar{w}(z,\bar{z})) \frac{\partial \bar{w}}{\partial z}\frac{\partial \bar{w}}{\partial \bar{z}} \,,$$

---

[1]See also [5] for previous work that studies holographic complexity using boundary sources.

with $T(z)$ and $\bar{T}(\bar{z})$ defined in (4).

Note that in (16) – which is a statement about operators – we have not included the contribution from the $T_{w\bar{w}}$ component. Let us briefly comment on why this is justified. It is well-known that classically, the trace of the energy-momentum tensor in a two-dimensional conformal field theory vanishes. In the quantum theory, $T_{w\bar{w}}$ no longer vanishes identically. However, since our calculation is performed in flat space, $T_{w\bar{w}}$ produces only contact terms when inserted in correlation functions. These contact terms do not contribute to correlation functions involving time-evolved operators. This can be seen directly from the definition of the time-evolution of an operator $\mathcal{O}$,

$$\mathcal{O}(t) = e^{\int_0^t d\tilde{t} H(\tilde{t})} \mathcal{O}(0) e^{-\int_0^t d\tilde{t} H(\tilde{t})}. \tag{17}$$

In correlation functions involving both $\mathcal{O}(0)$ and $H(\tilde{t})$, contact terms are proportional to $\delta(\tilde{t})$. Since $\tilde{t} = 0$ lies just outside of the integration range for $\tilde{t}$ in (17), the contribution of the contact terms drops out in the end. In fact, these contact term issues arise even in the ordinary treatment of conformal field theory on flat space using the time-slicing defined by the $w, \bar{w}$ coordinates. The textbook definition of the Hamiltonian in these coordinates is given by [17, 18]

$$H = L_0 + \bar{L}_0 = \int \frac{d\phi_w}{2\pi} (T(w) + \bar{T}(\bar{w})). \tag{18}$$

However, from the general expression (12), we see that even in these coordinates $T_{w\bar{w}}$ is in principle present in the Hamiltonian,

$$H = \int \frac{d\phi_w}{2\pi} (T(w) + \bar{T}(\bar{w}) + 2T_{w\bar{w}}(w, \bar{w})). \tag{19}$$

The arguments given above show that the trace part $T_{w\bar{w}}$ produces contact terms inside correlation functions that, however, do not contribute to time-evolution of operators. This explains why the textbook definition (18) is correct even though it differs from the expression obtained from (12).

Coming back to the derivation of the bulk dual to our circuit, combining (12) with (16) leads to the following expression for the Hamiltonian

$$H(t) = \int \frac{d\phi}{2\pi} \left[ \left( \left( \frac{\partial w}{\partial z} \right)^2 - \left( \frac{\partial w}{\partial \bar{z}} \right)^2 \right) T(w(z, \bar{z})) + \left( \left( \frac{\partial \bar{w}}{\partial \bar{z}} \right)^2 - \left( \frac{\partial \bar{w}}{\partial z} \right)^2 \right) \bar{T}(\bar{w}(z, \bar{z})) \right]. \tag{20}$$

Then, using a change of integration variable to rewrite the circuit generator as

$$Q(t) = \int \frac{d\phi}{2\pi} T(z) \epsilon(t, z) = -i \int \frac{d\phi}{2\pi} \partial_\phi w(z, \bar{z}) T(w(z, \bar{z})) \epsilon(t, w(z, \bar{z})), \tag{21}$$

we can read off $w(z, \bar{z})$ and $\bar{w}(z, \bar{z})$ from (13). In the remaining part of the section, we will come back to the two cases of the circuit starting from the case (b).

## 3.2 Realizing case (b)

Here, we find that the $w$ diffeomorphism is simply given by $f(t, z)$,

$$w(z, \bar{z}) = f(t, z), \tag{22}$$

where, following (15), $t = (z + \bar{z})/2$. On the other hand, the $\bar{w}$ diffeomorphism trivializes,

$$\bar{w}(z, \bar{z}) = \bar{z}. \tag{23}$$

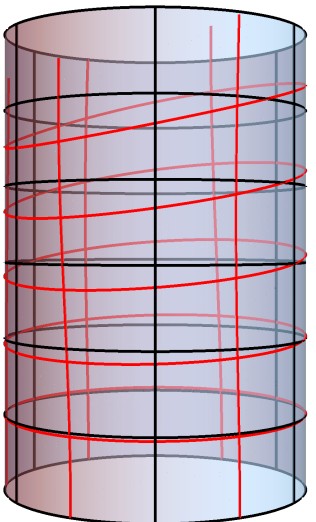

Figure 2: Flat cylinder in which the conformal field theory lives. Black curves correspond to slices of constant time (vertical) and angle (horizontal) associated with the $w, \bar{w}$ coordinates. The red curves represent constant time and angle associated with the $z, \bar{z}$ coordinates with the now infinitesimal diffeomorphism (22-23) specified by $f(t,z) = z + \varepsilon(3t^2 - 2t^3)\sin(z) + \mathcal{O}(\varepsilon^2)$ with $\varepsilon = 0.2$ (see also (40) for a definition of infinitesimal conformal transformations).

We do not implement any antiholomorphic conformal transformations, therefore the circuit only implements the trivial transformation $\bar{z} \to \bar{z}$ which leads to (23). An example of diffeomorphisms (22-23) and their effect on constant time slices is shown in figure 2. These diffeomorphisms lead to the following energy-momentum tensor expectation values,

$$\langle T_{zz} \rangle = -\frac{c}{24}\left(\frac{\partial w}{\partial z}\right)^2 = -\frac{c}{24}\frac{1}{4}(\dot{f}(t,z) + 2f'(t,z))^2,$$

$$\langle T_{z\bar{z}} \rangle = -\frac{c}{24}\left(\frac{\partial w}{\partial z}\right)\left(\frac{\partial w}{\partial \bar{z}}\right) = -\frac{c}{24}\frac{1}{4}(\dot{f}(t,z) + 2f'(t,z))\dot{f}(t,z), \tag{24}$$

$$\langle T_{\bar{z}\bar{z}} \rangle = -\frac{c}{24}\left(1 + \left(\frac{\partial w}{\partial \bar{z}}\right)^2\right) = -\frac{c}{24}\left(1 + \frac{1}{4}\dot{f}(t,z)^2\right),$$

in the background

$$ds^2_{(0)} = \left(\frac{1}{2}(\dot{f}(t,z) + 2f'(t,z))dz + \frac{1}{2}\dot{f}(t,z)d\bar{z}\right)d\bar{z}. \tag{25}$$

Note that this background metric is not of the form $dzd\bar{z}$, even after the circuit has reached the target state. In this region $t > t_{\text{final}}$, $\dot{f}(t,z) = 0$ and $ds^2_{(0)} = f'_{\text{final}}(z)dzd\bar{z}$. We may apply a Weyl transformation

$$ds^2_{(0)} \to e^{2\omega(z,\bar{z})}ds^2_{(0)} = \frac{1}{f'_{\text{final}}(F_{\text{final}}(f(t,z)))}ds^2_{(0)} \tag{26}$$

on top of this background to bring the metric to the form $dzd\bar{z}$ when $t > t_{\text{final}}$. Here $f_{\text{final}}$ is the total conformal transformation we produce after the circuit does its job and the inverse $F_{\text{final}}(z)$ is defined by $f_{\text{final}}(F_{\text{final}}(z)) = z$. At earlier times the metric has a more complicated form as one can see by comparing to (25), but it remains flat. In general, Weyl transformations change the Ricci scalar as

$$R \to e^{-2\omega}(R - 2\nabla_i\nabla^i\omega), \tag{27}$$

and thus lead to curved background metric. However, the Weyl transformation (26) we have chosen preserves $R = 0$. This can be seen from writing (27) in $w, \bar{w}$ coordinates,

$$e^{-2\omega}\partial_w\partial_{\bar{w}}\omega, \tag{28}$$

which vanishes for $\omega = \omega(w) + \bar{\omega}(\bar{w}) = \omega(f(t,z)) + \bar{\omega}(\bar{z})$. The energy-momentum tensor transforms under Weyl transformations as[2]

$$T_{ij} \rightarrow T_{ij} + \frac{c}{6}(\partial_i\omega\partial_j\omega - \frac{1}{2}g_{ij}\partial^k\omega\partial_k\omega - \nabla_i\nabla_j\omega + g_{ij}\nabla^k\nabla_k\omega). \tag{29}$$

Therefore, we find as expected for $t > t_{\text{final}}$

$$ds^2 = dz d\bar{z} \quad \text{and} \quad \langle T_{zz}\rangle = -\frac{c}{24}f'_{\text{final}}(z)^2 + \frac{c}{12}\{f_{\text{final}}(z),z\}, \quad \langle T_{z\bar{z}}\rangle = 0, \quad \langle T_{\bar{z}\bar{z}}\rangle = -\frac{c}{24}. \tag{30}$$

The intermediate form of the energy-momentum tensor for $t_{\text{initial}} < t < t_{\text{final}}$ depends on the particular Weyl-rescaling we do and can be found simply by using the transformation rule (29).

Note that the Hamiltonian is not invariant under Weyl transformations due to the energy-momentum tensor transformation (29). However, the additional term in the Hamiltonian is proportional to the identity operator and has no observable effect.

Let us briefly discuss uniqueness of the circuit we have constructed. The circuit and its bulk dual is specified by the boundary metric and energy-momentum tensor expectation value. Therefore, one might ask what is the correct choice of these quantities to implement the same sequence of states as in section 2 – equations (24) and (25) on their own, or supplemented with the Weyl rescaling (26)? The answer is that these two choices are equivalent implementations of the same circuit. Because the Hamiltonian changes trivially under the Weyl transformation (26), this transformation does not affect the sequence of states in the circuit. What changes, however, are the expectation values of the energy-momentum tensor. This feature is special to $T_{ij}$, general tensor fields are invariant under Weyl transformations. But because the energy-momentum tensor depends directly on the background metric through Weyl anomaly and conservation equations, its expectation values are comparable only if they are evaluated in the same background. In other words, the Hilbert space operator defined by $T_{ij}$ in the background $ds^2_{(0)}$ differs from the Hilbert space operator defined by $T_{ij}$ in the background $e^{2\omega}ds^2_{(0)}$. The Weyl transformation (26) we have chosen merely puts the metric at $t > t_{\text{final}}$ in the same form as that used in section 2 so that we can compare the expectation values $\langle T_{ij}\rangle$ in the circuit from section 2 and its reformulation in this section. As expected, once we transform to the background $ds^2_{(0)} = dz d\bar{z}$, we find agreement with the expectation values from section 2.

### 3.3 Another look at the circuit from case (a)

As we have discussed earlier, the natural interpretation of the circuit in case (a) is that of a sequence of states in different realizations of considered conformal field theory, i.e. living in different spacetimes. However, the realization of case (b) provides us with a possibility of an alternative perspective on case (a). In fact, as we will see, the two cases can be realized in a very similar manner upon identifying $\tau = t$.

An additional issue to take into account is that in case (a) we need to perform a slight reformulation of the circuit in order to be able to demand equality of $H(t)$ given by (20) and $Q(t)$ specified in (21). The reason for this is that for a trivial conformal transformation

---

[2]This equation can be derived as a statement for the expectation value of the energy-momentum tensor from the Weyl anomaly equation. One may check that this also holds as a operator statement by comparing with the two-point function of the energy-momentum tensor in a general background. We have done this perturbatively up to second order (included) in perturbation theory around flat space.

$f(t,z) = z$ the circuit generator $Q_{(a)}(\tau = t)$ from section 2 vanishes while we want the Hamiltonian $H(t)$ to reduce to the standard time evolution in a conformal field theory governed by $H(t) = H_0 = L_0 + \bar{L}_0$. Therefore, we introduce a modification of $Q_{(a)}(t)$ by adding $H_0$, $Q_{(a)}(t) \rightarrow Q_{(a)}(t) + H_0$ before identifying it with $H(t)$. This modification does not change the energy-momentum tensor expectation value[3] and only leads to an additional unobservable phase if the reference state is a primary state such as the vacuum state $|0\rangle$ that we are using as reference state. Therefore, this modification does not change the physics of the problem at hand.

Then, using (20) and (21) we find that the $\bar{w}$ diffeomorphism trivializes again, $\bar{w}(z, \bar{z}) = \bar{z}$, while the $w$ diffeomorphism satisfies

$$\dot{w}(t, \phi) = 1 + \epsilon(t, w(t, \phi)). \tag{31}$$

We may rewrite (31) by using the definition of $\epsilon$ in (5) and introducing inverse functions $W$ and $F$ defined by

$$w(t, W(t, \phi)) = \phi, \quad f(t, F(t, z)) = z, \tag{32}$$

giving[4]

$$-\frac{\dot{W}(t, \phi)}{W'(t, \phi)} = 1 - \frac{\dot{F}(t, t + i\phi)}{F'(t, t + i\phi)}. \tag{33}$$

It is then easy to see that case (a) and (b) are implemented by sources $g_{ij}^{(0)}$ described by closely related diffeomorphisms $w(z, \bar{z})$ differing only in a total vs. partial derivative with respect to the physical time $t$ in their defining equations.

Applying again the Weyl transformation (26), we find the following energy-momentum tensor expectation values for $t > t_{\text{final}}$,

$$\langle T_{zz} \rangle = -\frac{c}{24} + \frac{c}{12}\{f_{\text{final}}(z), z\}, \quad \langle T_{z\bar{z}} \rangle = 0, \quad \langle T_{\bar{z}\bar{z}} \rangle = -\frac{c}{24}. \tag{34}$$

Compared to the well-known transformation law of the energy-momentum tensor under conformal transformations,

$$T(z) \rightarrow f'(z)^2 T(z) + \frac{c}{12}\{f(z), z\}, \tag{35}$$

we find that in this circuit the $f'(z)^2$ prefactor is absent in the final value of the energy-momentum tensor expectation value. Hence, we conclude that the circuit (b) more faithfully implements gradual conformal transformations in the sense that the final state yields the well-known energy-momentum tensor transformation rule. Nevertheless, the circuit (a) possesses interesting features with regard to holographic complexity proposals, as we explain in section 5 and thus deserves to be studied in detail.

## 4 Mapping to gravity

The results of section 3 correspond to a path-integral prescription within quantum field theory for defining the circuits of section 2 in terms of evolution in physical time. The key outcomes of this analysis are that the circuits are defined in flat space and that it is a particular time foliation of flat space that triggers, as time progresses, the transformation of interest.

---

[3]The modification is equivalent to the replacement $f \rightarrow f + \text{const.}$ in (9). If $\langle T(z) \rangle$ is constant, this does not change the energy-momentum expectation value.

[4]Note that $\dot{F}(t, z)$ denotes a derivative of $F$ w.r.t. its first argument and not a total derivative w.r.t. $t$. Likewise, $F'(t, z)$ is a derivative w.r.t. the second argument of the function.

The holographic dictionary associates the metric underlying the path-integral formulation with the metric on the asymptotic boundary and the corresponding energy-momentum tensor with the subleading fall-off of the bulk metric [19]. Usually, the boundary metric and the boundary energy-momentum tensor, even if known over the entire boundary, do not specify the dual geometry in a closed form. However, in three bulk dimensions, which is the situation of interest, the Fefferman-Graham near-boundary expansion of the bulk metric truncates and the input we provide from section 3 does specify the full bulk metric in a closed form.

To be more specific, if $g_{ij}^{(0)}$ denotes the boundary metric and $\langle T_{ij} \rangle$ the allowed expectation value of the energy-momentum tensor, the exact gravity dual to the corresponding time evolution of a state in a holographic conformal field theory takes the form [19]

$$ds^2 = \frac{dr^2}{r^2} + \left( \frac{1}{r^2} g_{ij}^{(0)} + g_{ij}^{(2)} + r^2 g_{ij}^{(4)} \right) dx^i dx^j , \tag{36}$$

where $r$ is the radial direction with the asymptotic boundary at $r = 0$ and

$$g_{ij}^{(2)} = -\frac{1}{2} R^{(0)} g_{ij}^{(0)} - \frac{6}{c} \langle T_{ij} \rangle \quad \text{and} \quad g_{ij}^{(4)} = \frac{1}{4} (g^{(2)} (g^{(0)})^{-1} g^{(2)})_{ij} . \tag{37}$$

Therefore, the gravity dual to the circuits of interest is obtained by inserting into the above expression the form of the boundary metric and the associated expectation value of the energy-momentum tensor discussed in the previous section. Concretely, for the circuit (b) the boundary metric is given by (26) and the energy-momentum tensor expectation value is determined from (24) and (29) and analogously for the circuit (a). The results then basically tell us which time-slicing of pure $AdS_3$ one has to choose in order to implement the circuit of interest that acts on the vacuum state.

The derived bulk metric forms a possible basis for first-principle derivations of bulk duals to various field theory cost functions which have been proposed previously [5,9,11–13]. It can also provide conformal field theory insights on conjectured bulk complexity measures such as "complexity=volume" [20], "complexity=action" [21], "complexity=volume 2.0" [22], or the infinite class of complexity measures recently proposed in [23].

## 5 Lessons for holographic complexity

Having derived the bulk dual to our circuit, we now turn to the study of bulk duals of boundary cost functions and – vice versa – boundary duals to bulk complexity measures. To be specific, we will here concentrate on two simple examples: the "complexity=volume" proposal [20] and the squared Fubini-Study cost and associated complexity [24] applied in this context in [12,13].

Let us make sure that all the readers are on the same page and discuss briefly what we mean by the squared Fubini-Study cost and associated complexity. The total cost of a circuit is a non-negative number assigned in a systematic way to each of its layers and integrated over the circuit parameter. The discussion of costs in the high-energy physics literature is based on [24–26]. The Fubini-Study cost is the one that originates from the distance that the circuit traverses in the Hilbert space when acting on a given state. The associated complexity arises from minimization of the total cost (distance). In the notation that we adopted in section 2 and following [12,13], the complexity is given by

$$C_{\text{FS}} = \min \int d\tau \, F_{\text{FS}}(\tau)^2 , \tag{38}$$

where

$$F_{\text{FS}}(\tau) = \sqrt{\langle 0 | U^\dagger(\tau) Q^\dagger(\tau) Q(\tau) U(\tau) | 0 \rangle - |\langle 0 | U^\dagger(\tau) Q(\tau) U(\tau) | 0 \rangle|^2} . \tag{39}$$

Note especially the inclusion of the square in equation (38). This means that compared to the problem of geodesic motion in curved spaces, the functional that we are working with is more similar to the kinetic energy than the length functional. An earlier critical investigation of this and similar cost functionals can be found in [27]. When trying to make contact with the complexity=volume proposal via a Fubini-Study-based ansatz, including the square is important because we know that volume scales linearly in the central charge $c$, and so should complexity. Also, as shown in [5,6], the complexity (38) matches the change of volume under infinitesimal local conformal transformations to the leading nontrivial order. We will hence further study this cost further in this section, and allowing ourselves a little imprecision of nomenclature, we will refer to (38) as Fubini-Study complexity.

When the conformal transformation is expressed as a perturbative series around the identity,

$$f(t,z) = z + \varepsilon f_1(t,z) + \varepsilon^2 f_2(t,z) + \mathcal{O}(\varepsilon^3), \tag{40}$$

the Fubini-Study complexity and the result of the "complexity=volume" calculation in the relevant Bañados geometry are known to be related at the order $\varepsilon^2$ [5,6]. To be more specific, Ref. [6] considered the gravity dual to the state corresponding to $z \to f_{\text{final}}(z)$ and in this state calculated the "complexity=volume" proposal in the expansion in $\varepsilon$, which was found to be related to the Fubini-Study complexity measure in [5]. Let us revisit this calculation but now at all time instances in the circuit. For $t > t_{\text{final}}$, this reduces to the setup of [6].

For the case (b), we find the following change in volume compared to the vacuum state in appendix A [5],

$$V_{(b)} - V_{\text{pure AdS}_3} = \varepsilon^2 \frac{\pi}{4} \sum_n (|n|^3 - |n|) f_1^n(t) f_1^{-n}(t) \tag{41}$$

$$+ \varepsilon^3 \frac{\pi}{4} \sum_n (|n|^3 - |n|) \left( 2 f_1^n(t) f_2^{-n}(t) - i \sum_m m f_1^n(t) f_1^m(t) f_1^{-n-m}(t) \right) + \mathcal{O}(\varepsilon^4).$$

On the other hand, for case (a) we cannot give a general answer for the volume of extremal slices because (31) cannot be solved for arbitrary time dependence. The most interesting special case is the one in which the time-dependence equals that of the optimal path in the Fubini-Study complexity functional of [12,13]. In this case, we obtain[6]

$$V_{(a)} - V_{\text{pure AdS}_3} = \varepsilon^2 \frac{\pi}{4} \sum_n (|n|^3 - |n|) \frac{f_1^n(t) f_1^{-n}(t)}{n^2} + \mathcal{O}(\varepsilon^3). \tag{42}$$

See appendix A for details, including the third and the fourth order contributions to (42). We can think about the difference $V_{(a,b)} - V_{\text{pure AdS}_3}$ as a notion of complexity of formation, i.e. in our case a way of assigning a cost of transforming the vacuum state into the state at $t_{\text{final}}$. Such a notion was considered earlier in the case of thermofield double states in [28].

It is instructive to compare the above results to the Fubini-Study complexity of [12,13]. We derive general expressions for this complexity functional to fourth order in perturbation theory in appendix B. Interestingly, the volume change (41) in the circuit (b) matches[7] as far

---

[5]The results in appendix A are given in terms of parameters $C_1^n$, $C_2^n$, etc. which are the $n$-th Fourier modes parametrizing the location of the time slice on the boundary in $w$, $\bar{w}$ coordinates expanded in perturbation theory. Thus, these parameters are obtained directly from (31), taking into account that the $C_1^n$, $C_2^n$ parameters are Fourier modes w.r.t. $\phi_w = (w - \bar{w})/(2i)$. Note also that the calculation in appendix A is performed in Lorentzian signature. Finally, note that the Fourier modes of $f_1, f_2$ satisfy $(f_{1,2}^n)^* = f_{1,2}^{-n}$ such that the final expression for the volume is real despite the presence of the imaginary unit in (41).

[6]See section B for the derivation of the optimal path.

[7]Up to a prefactor which is undetermined in the complexity functional anyway.

as the third order in $\varepsilon$,

$$
\begin{aligned}
\Rightarrow C_{\text{FS}} = {} & \varepsilon^2 \sum_n \frac{c}{24}(|n|^3 - |n|) f_1^n(t) f_1^{-n}(t) \\
& + \varepsilon^3 \sum_n \frac{c}{24}(|n|^3 - |n|) \left[ 2 f_1^n(t) f_2^{-n}(t) - i \sum_m m f_1^n(t) f_1^m(t) f_1^{-n-m}(t) \right] + \mathcal{O}(\varepsilon^4),
\end{aligned}
\tag{43}
$$

but disagrees in the fourth order (see (74) and (89)). The Fubini-Study complexities in the circuits (a) and (b) are equal to each other up to the third order in perturbation theory, therefore (43) holds for both circuits. These results show that the Fubini-Study distance is not directly related to volume changes. This rules out this possibility put forward in [5]. Note that generalizations of $C_{\text{FS}}$ obtained by counting the cost in the circuit as some power of the Fubini-Study metric (a procedure that does not change the optimal path in the circuit) cannot match the volume change[8] since also in this case, the change in the maximal volume disagrees with the Fubini-Study complexity (see appendix B).

The bulk dual to the circuits we have derived in secs. 3 and 4 allows – at least in principle – a derivation of bulk duals to cost functions such as the Fubini-Study metric from first principles. The Fubini-Study metric is related to a connected two-point function of the Hamiltonian. In general, connected two-point functions of the energy-momentum tensor are obtained from the boundary perspective by applying variations w.r.t. the boundary metric onto the one-point function,

$$
\langle T_{ij} T_{kl} \rangle = \frac{2}{\sqrt{g_{(0)}}} \frac{\delta}{\delta g_{(0)}^{ij}} \langle T_{kl} \rangle.
\tag{44}
$$

In this way, the two-point function of the Hamiltonian entering the Fubini-Study complexity definition (77) can be derived. The important point is now, that using the relation between the energy-momentum tensor one-point function and the bulk metric in (37), we may translate this into a bulk calculation giving the same two-point function. This allows in principle writing down the gravity dual to the Fubini-Study cost function used in [12, 13]. Of course, similar derivations work for other cost functions. Our method allows for deriving bulk duals to any cost function defined from energy-momentum tensor correlators or vice versa boundary duals to bulk cost functions defined as functionals of the bulk metric. It may of course be the case that the bulk duals for such cost functions does not reduce to a simple geometric quantity in the bulk. Indeed, in general energy-momentum tensor correlators are derived by applying variations which necessarily change the bulk metric (although of course only slightly) and lead us to different bulk geometry. Therefore, we expect to find simple geometric duals only for certain special cases in which the effect of the variation of the background drops out in the end. This is also reminiscent to the situation with entanglement entropy, represented as a property of a given bulk geometry [29–32], and general Renyi entropies requiring backreaction [33]. We leave this topic for further research.

## 6 Summary and outlook

We have derived gravity duals to circuits generating conformal transformations in the boundary conformal field theory. Our construction was based on identifying the circuit generator $Q(\tau)$ with the physical Hamiltonian $H(t)$ generating time evolution in a specific background metric $g_{ij}^{(0)}$ on the boundary. Therefore, we identify the auxiliary circuit parameter $\tau$ with the physical time $t$. This is the main new feature of our construction compared to previous work on holographic complexity. Furthermore, the identification of the circuit parameter with

---

[8]We would like to thank Alex Belin for bringing this possibility to our attention.

the physical time also allows us to derive a bulk dual to the entire circuit using the Fefferman-Graham expansion [34]. Finally, we studied relations between "complexity=volume" [20] and the Fubini-Study complexity measure proposed in [12,13]. As a byproduct of this analysis, we managed to rule out the possibility that this complexity measure and the "complexity=volume" proposal are directly related [5].

The construction of precise gravity duals to quantum circuits presented in this paper provides a new setting to study field theory cost functions directly in the bulk or conversely to derive the dual boundary quantities associated to bulk observables like the change in the volume of an extremal time slice under time evolution. Furthermore, another interesting question is whether any of the previously studied cost functions in [9–13] can be mapped to geometric quantities in the bulk. For instance, in [8] it was demonstrated that the Fubini-Study distance on the space of circuits starting from scalar primary states is encoded in the maximal and minimal perpendicular distances between infinitesimally close timelike geodesics in AdS. Conversely, it would be interesting to understand better bulk candidates for costs considered in [35–38]. To make further progress in this direction, cost functions on the boundary have to be determined in terms of the bulk metric or conversely bulk observables in terms of conformal field theory quantities like the boundary energy-momentum tensor. Our construction allows such derivations using directly the holographic dictionary. One interesting clue that one can use in this quest is that the costs associated with our circuits should be UV-finite and, therefore, should not directly stem from bulk objects extending all the way to the asymptotic boundary. The reason for the finiteness is that we do not need to alter the entanglement structure of the reference state at arbitrarily short scales, but only in the IR.

Furthermore, the approach presented here may be generalized in a number of ways. A simple generalization is to allow Virasoro generators from the two copies of the Virasoro algebra to act simultaneously in the circuit. Because both copies decouple, this is an obvious generalization of our results from section 3. Another, more interesting generalization is possible by allowing the boundary metric to be curved. This allows for a circuit construction where the reference and target state remain the same as here, while the sequence of states interpolating between them changes. In general, such constructions are more difficult to interpret in terms of gates acting on states, and hence the precise sequence of states between the reference and target state is harder to derive. One possibility to construct such a geometry is to consider a general Weyl-rescaled geometry without restricting the Weyl factor to allow only flat boundary metrics, as we did here. For such a circuit, it is possible to construct a one-norm cost function similar to that considered in [9]. We will discuss such circuits in more detail in an upcoming work.

Finally, it would be very interesting to make further contact with the approach to gravity duals of circuits pioneered in [5]. In particular, both approaches use non-trivial boundary metrics to define circuits. It would be certainly interesting to understand possible relations between them with a hope to advance in this way the field of gravity duals to cost functions and complexity from a quantitative perspective underlying the present work.

## Acknowledgements

We are grateful to Souvik Banerjee, Alex Belin, Blagoje Oblak, René Meyer and Leo Shaposhnik for useful discussions. The Gravity, Quantum Fields and Information group at AEI was supported by the Alexander von Humboldt Foundation and the Federal Ministry for Education and Research through the Sofja Kovalevskaja Award. The work of M. F. is supported through the grants CEX2020-001007-S and PGC2018-095976-B-C21, funded by MCIN/AEI/10.13039/501100011033 and by ERDF A way of making Europe. The work of A.-L. W. is supported by

DFG, grant ER 301/8-1 | ME 5047/2-1. The work M. G. is supported by Germany's Excellence Strategy through the Würzburg-Dresden Cluster of Excellence on Complexity and Topology in Quantum Matter - ct.qmat (EXC 2147, project-id 390858490).

# A  Maximal volume slices

In this section, we compute the volume of extremal slices in bulk geometries obtained from diffeomorphisms of pure AdS$_3$. In other words, we calculate the proposed bulk dual to complexity in the "complexity=volume" approach [20] in these geometries. The extremal slices we consider asymptote to a constant time slice on the boundary in $z, \bar{z}$ coordinates. Equivalently, in $w, \bar{w}$ coordinates the bulk metric is the standard pure AdS$_3$ metric while the slice asymptotes to a diffeomorphism of the constant time slice in $z, \bar{z}$ coordinates. The calculation is done perturbatively to fourth order in the perturbation parameter.

We start with the global AdS$_3$ metric in coordinates

$$ds^2 = -\cosh^2\rho \, dt^2 + d\rho^2 + \sinh^2\rho \, d\phi^2. \tag{45}$$

The embedding of the maximal volume slice is determined by $t(\phi, \rho)$. The induced metric on the maximal volume slice is given by

$$ds_{\text{ind.}}^2 = \left(1 - \cosh^2\rho \left(\frac{\partial t}{\partial \rho}\right)^2\right) d\rho^2 - 2\cosh^2\rho \frac{\partial t}{\partial \rho} \frac{\partial t}{\partial \phi} d\rho d\phi \tag{46}$$

$$+ \left(\sinh^2\rho - \cosh^2\rho \left(\frac{\partial t}{\partial \phi}\right)^2\right) d\phi^2. \tag{47}$$

For the zeroth order in perturbation theory the boundary conditions are $t(\phi, \rho \to \infty) = t_0 = \text{const.}$ and the maximal volume slice is a constant time slice $t(\phi, \rho) = t_0$. The volume is given as the square root of the determinant $\gamma$ of the induced metric, giving a UV divergent result

$$V_{(0)} = \int_0^{1/\epsilon_{\text{UV}}} d\rho \int_0^{2\pi} d\phi \sqrt{\gamma} = \int_0^{1/\epsilon_{\text{UV}}} d\rho \int_0^{2\pi} d\phi \sinh\rho = 2\pi\left(\frac{1}{\epsilon_{\text{UV}}} - 1\right). \tag{48}$$

**First and second order:**  We now expand around the zeroth order solution with expansion parameter $\varepsilon$,

$$t(\phi, \rho) = t_0 + \varepsilon t_1(\phi, \rho) + \varepsilon^2 t_2(\phi, \rho) + \dots. \tag{49}$$

Up to second order the square root of the determinant of the induced metric is given by

$$\sqrt{\gamma} = -\frac{\left(\cosh^2\rho \sinh^2\rho \, \dot{t}_1^2 + \cosh^2\rho \, t_1'^2\right)\varepsilon^2}{2\sinh\rho} + \sinh\rho, \tag{50}$$

where $\dot{t}_1 = \frac{\partial}{\partial \rho} t_1(\rho, \phi)$ and $t_1' = \frac{\partial}{\partial \phi} t_1(\rho, \phi)$. Note that the first order term $\mathcal{O}(\varepsilon)$ in $\sqrt{\gamma}$ vanishes and hence the volume of the extremal slice to first order is equal to the zeroth order result. To determine the location of the extremal volume slice to first order, we perform a variation with respect to $t_1$, giving

$$-\left(3\cosh^3\rho - 2\cosh\rho\right)\sinh\rho \, \dot{t}_1 - \cosh^2\rho \, t_1'' - \cosh^2\rho \sinh^2\rho \, \ddot{t}_1 = 0. \tag{51}$$

Decomposing $t_1$ in a Fourier series, $t_1 = \sum_n t_1^n(\rho)e^{in\phi}$, yields

$$n^2\cosh^2\rho \, t_1^n(\rho) - \left(3\cosh^3\rho - 2\cosh\rho\right)\sinh\rho \frac{\partial}{\partial \rho} t_1^n(\rho) - \cosh^2\rho \sinh^2\rho \frac{\partial^2}{(\partial \rho)^2} t_1^n(\rho) = 0. \tag{52}$$

The general solution is given as a sum of two linearly independent solutions

$$t_1^n(\rho) = C_{n,+} t_{1,+}^n(\rho) + C_{n,-} t_{1,-}^n(\rho), \tag{53}$$

where

$$t_{1,\pm}^n(\rho) = \left(\frac{\cosh(\rho) - 1}{\cosh(\rho) + 1}\right)^{\pm |n|/2} \frac{\cosh(\rho) \pm |n|}{\cosh(\rho)}. \tag{54}$$

However, $\lim_{\rho \to 0} t_{1,-}^n = \infty$ which is not consistent with the perturbative expansion. Therefore, the solution is restricted to

$$t_1^n(\rho) = C_1^n \left(\frac{\cosh(\rho) - 1}{\cosh(\rho) + 1}\right)^{|n|/2} \frac{\cosh(\rho) + |n|}{\cosh(\rho)}. \tag{55}$$

The constant $C_1^n$ is determined from the boundary conditions. Inserting this into (50) yields the following volume of the extremal slice to second order in the perturbation expansion,

$$
\begin{aligned}
V &= \int_0^{1/\epsilon_{\text{UV}}} d\rho \int_0^{2\pi} d\phi \sqrt{\gamma} \\
&= V_{(0)} - \varepsilon^2 \pi \int_0^{1/\epsilon_{\text{UV}}} d\rho \frac{\cosh^2 \rho}{\sinh \rho} \sum_n \left(n^2 t_1^n t_1^{-n} + \sinh^2 \rho \frac{\partial t_1^n}{\partial \rho} \frac{\partial t_1^{-n}}{\partial \rho}\right) \\
&= V_{(0)} + \varepsilon^2 \pi \sum_n \left(-\frac{n^2}{\epsilon_{\text{UV}}} + |n|^3 - |n|\right) C_1^n C_1^{-n}.
\end{aligned} \tag{56}
$$

Taking into account that the cutoff surface $\rho = 1/\epsilon_{\text{UV}}$ also changes under the diffeomorphism $w(z, \bar{z})$, the non-universal cutoff dependent terms in second order in the perturbation parameter $\varepsilon$ cancel. Therefore, we finally obtain a finite result for the change in volume of the extremal slice compared to pure AdS$_3$,

$$V_{(2)} = V|_{\mathcal{O}(\varepsilon^2)} = \pi \sum_n \left(|n|^3 - |n|\right) C_1^n C_1^{-n}. \tag{57}$$

**Third order:** The third order term $V_{(3)}$ is derived in the same way as the second order one. To third order in $\varepsilon$, the determinant of the induced metric reads

$$\sqrt{\gamma}|_{\mathcal{O}(\varepsilon^3)} = -\frac{\cosh^2 \rho \sinh^2 \rho \, \dot{t}_1 \dot{t}_2 + \cosh^2 \rho \, t_1' t_2'}{\sinh \rho}. \tag{58}$$

The equation of motion for $t_2$ is the same as the one for $t_1$. Thus, the (UV cutoff independent) change in volume to third order is given by

$$V_{(3)} = 2\pi \sum_n \left(|n|^3 - |n|\right) C_1^n C_2^{-n}. \tag{59}$$

**Fourth order:** In this order, the determinant of the induced metric is given by

$$
\begin{aligned}
\sqrt{\gamma}|_{\mathcal{O}(\varepsilon^4)} = -\frac{1}{8 \sinh^3 \rho} \Big[ & \cosh^4 \rho \sinh^4 \rho \, \dot{t}_1^4 + 2 \cosh^4 \rho \sinh^2 \rho \, \dot{t}_1^2 t_1'^2 + \cosh^4 \rho \, t_1'^4 \\
& + 4 \cosh^2 \rho \sinh^4 \rho \, \dot{t}_2^2 + 4 \cosh^2 \rho \sinh^2 \rho \, t_2'^2 \\
& + 8 \cosh^2 \rho \sinh^4 \rho \, \dot{t}_1 \dot{t}_3 + 8 \cosh^2 \rho \sinh^2 \rho \, t_1' t_3' \Big].
\end{aligned} \tag{60}
$$

This gives the following equation of motion for $t_3$,

$$
\begin{aligned}
&\cosh^4 \rho \sinh^2 \rho \, \dot{t}_1^2 t_1'' + 3\cosh^4 \rho \, t_1'^2 t_1'' + \left(\cosh^5 \rho \sinh^3 \rho + 4\cosh^3 \rho \sinh^5 \rho\right)\dot{t}_1^3 \\
&+ 4\cosh^4 \rho \sinh^2 \rho \, \dot{t}_1' t_1' \dot{t}_1 - \left(\cosh^5 \rho \sinh \rho - 4\cosh^3 \rho \sinh^3 \rho\right)t_1'^2 \dot{t}_1 \\
&+ 3\cosh^4 \rho \sinh^4 \rho \, \dot{t}_1^2 \ddot{t}_1 + \cosh^4 \rho \sinh^2 \rho \, t_1'^2 \ddot{t}_1 \\
&+ 2\cosh^2 \rho \sinh^4 \rho \, \ddot{t}_3 + 2\cosh^2 \rho \sinh^2 \rho \, t_3'' + 2\left(\cosh^3 \rho \sinh^3 \rho + 2\cosh \rho \sinh^5 \rho\right)\dot{t}_3 = 0 .
\end{aligned}
\tag{61}
$$

This can be slightly simplified by inserting the equation of motion for $t_1$,

$$
\begin{aligned}
&\cosh^4 \rho \, t_1'^2 t_1'' + \cosh^3 \rho \sinh^5 \rho \, \dot{t}_1^3 + 2\cosh^4 \rho \sinh^2 \rho \, \dot{t}_1' t_1' \dot{t}_1 - \cosh^3 \rho \sinh \rho \, t_1'^2 \dot{t}_1 \\
&+ \cosh^4 \rho \sinh^4 \rho \, \dot{t}_1^2 \ddot{t}_1 + \sinh^2 \rho \left(\cosh^2 \rho \sinh^2 \rho \, \ddot{t}_3 + \cosh^2 \rho \, t_3''\right. \\
&\hspace{5cm} \left. + \sinh \rho \cosh \rho (3\cosh^2 \rho - 2)\dot{t}_3\right) = 0 .
\end{aligned}
\tag{62}
$$

Decomposing $t_3$ in a Fourier series gives

$$
\cosh^2 \rho \sinh^2 \rho \, \ddot{t}_3^n + \sinh \rho \cosh \rho (3\cosh^2 \rho - 2)\dot{t}_3^n - n^2 \cosh^2 \rho \, t_3^n = g_n(\rho) ,
\tag{63}
$$

where we have put all the $t_1$-dependent parts into the function $g_n(\rho)$. The solution to this inhomogenous differential equation is given by a sum of a special inhomogenous solution and the solution of the homogenous equation with $g_n(\rho) = 0$. Since the homogenous equation is equivalent to the e.o.m. for $t_1^m$ and $t_2^m$, the solution is already known. The inhomogenous solution can be obtained by a Greens function ansatz:

$$
-n^2 \cosh^2 \rho \, G(\rho, \rho_0) + \cosh \rho \sinh \rho (3\cosh^2 \rho - 2)\frac{\partial}{\partial \rho}G(\rho, \rho_0)
\tag{64}
$$

$$
+ \cosh^2 \rho \sinh^2 \rho \frac{\partial^2}{\partial \rho^2}G(\rho, \rho_0) = \delta(\rho - \rho_0) .
\tag{65}
$$

It is clear that the solution of (65) is equal to the solution of (52) when $\rho \neq \rho_0$, therefore we make the ansatz

$$
G(\rho, \rho_0) = \begin{cases} C_+ t_{1,+}^n + C_- t_{1,-}^n , & \rho < \rho_0 , \\ \hat{C}_+ t_{1,+}^n + \hat{C}_- t_{1,-}^n , & \rho > \rho_0 . \end{cases}
\tag{66}
$$

Requiring continuity of $G(\rho, \rho_0)$ at $\rho = \rho_0$ and the proper discontinuity of its derivative to reproduce the right hand side of (65) fixes the coefficients $C_\pm$ and $\hat{C}_\pm$. Integrating over $\rho_0$ we obtain

$$
\begin{aligned}
t_{3,\text{inhom.}}^n(\rho) =& \frac{t_{1,+}^n(\rho)}{2|n|(|n|^2 - 1)}\left[-\int_0^\infty d\rho_0 \frac{(\cosh \rho_0 + |n|)\tanh(\rho_0/2)^{|n|}}{\sinh \rho_0 \cosh \rho_0}g_n(\rho_0) \right. \\
&\left. + \int_\rho^\infty d\rho_0 \frac{(\cosh \rho_0 - |n|)\tanh(\rho_0/2)^{-|n|}}{\sinh \rho_0 \cosh \rho_0}g_n(\rho_0)\right] \\
&+ \frac{t_{1,-}^n(\rho)}{2|n|(|n|^2 - 1)}\int_0^\rho d\rho_0 \frac{(\cosh \rho_0 + |n|)\tanh(\rho_0/2)^{|n|}}{\sinh \rho_0 \cosh \rho_0}g_n(\rho_0) .
\end{aligned}
\tag{67}
$$

The inhomogenous part $t_{3,\text{inhom.}}^n$ of the solution vanishes at $\rho = 0, \infty$:

$$
\begin{aligned}
\lim_{\rho \to \infty} t_{3,\text{inhom.}}^n(\rho) &= \left(\int_0^\infty d\rho_0 \frac{t_{1,+}^n(\rho_0)g_n(\rho_0)}{2|n|(|n|^2 - 1)\sinh \rho_0}\right)(t_{1,+}^n(\rho \to \infty) - t_{1,-}^n(\rho \to \infty)) = 0 , \\
\lim_{\rho \to 0} t_{3,\text{inhom.}}^n(\rho) &= \left(\int_0^\infty d\rho_0 \frac{(t_{1,-}^n(\rho_0) - t_{1,+}^n(\rho_0))g_n(\rho_0)}{2|n|(|n|^2 - 1)\sinh \rho_0}\right)t_{1,+}^n(\rho \to 0) = 0 .
\end{aligned}
\tag{68}
$$

Therefore, to impose the boundary conditions obeyed by $t_3$ we only need to consider the homogenous part of the solution as before. Furthermore, it can be shown that the inhomogenous part $t_{3,\text{inhom.}}^n$ does not contribute to the volume change. The contribution of $t_{3,\text{inhom.}}^n$ to $V_{(4)}$ is proportional to

$$
\int d\rho \frac{\cosh^2 \rho}{\sinh \rho}(\sinh^2 \rho \, \dot{t}_{1,+}^n(\rho)\dot{t}_{3,\text{inhom.}}^{-n}(\rho) + n^2 t_{1,+}^n(\rho)t_{3,\text{inhom.}}^{-n}(\rho))
$$

$$
= -\int_0^\infty d\rho \frac{\cosh^2 \rho}{\sinh \rho}(\sinh^2 \rho \, \dot{t}_{1,+}^n(\rho)\dot{t}_{1,+}^{-n}(\rho) + n^2 t_{1,+}^n(\rho)t_{1,+}^{-n}(\rho)) \int_0^\infty d\rho_0 \frac{t_{1,+}^{-n}(\rho_0)g_{-n}(\rho_0)}{\sinh \rho_0}
$$

$$
+ \int_0^\infty d\rho \frac{\cosh^2 \rho}{\sinh \rho}(\sinh^2 \rho \, \dot{t}_{1,+}^n(\rho)\dot{t}_{1,+}^{-n}(\rho) + n^2 t_{1,+}^n(\rho)t_{1,+}^{-n}(\rho)) \int_\rho^\infty d\rho_0 \frac{t_{1,-}^{-n}(\rho_0)g_{-n}(\rho_0)}{\sinh \rho_0}
$$

$$
+ \int_0^\infty d\rho \frac{\cosh^2 \rho}{\sinh \rho}(\sinh^2 \rho \, \dot{t}_{1,+}^n(\rho)\dot{t}_{1,-}^{-n}(\rho) + n^2 t_{1,+}^n(\rho)t_{1,-}^{-n}(\rho)) \int_0^\rho d\rho_0 \frac{t_{1,+}^{-n}(\rho_0)g_{-n}(\rho_0)}{\sinh \rho_0} .
$$

(69)

Using that

$$
\int d\rho \frac{\cosh^2 \rho}{\sinh \rho}(\sinh^2 \rho \, \dot{t}_{1,+}^n \dot{t}_{1,-}^{-n} + n^2 t_{1,+}^n t_{1,-}^{-n})
$$

(70)

$$
= |n|^2 \left( \frac{1}{|n|} - |n| + \cosh \rho + \frac{1}{\cosh \rho} \right)(\tanh(\rho/2))^{2|n|} ,
$$

(71)

and

$$
\int d\rho \frac{\cosh^2 \rho}{\sinh \rho}(\sinh^2 \rho \, \dot{t}_{1,+}^n \dot{t}_{1,-}^{-n} + n^2 t_{1,+}^n t_{1,-}^{-n}) = |n|^2 \left( \cosh \rho - \frac{1}{\cosh \rho} \right),
$$

(72)

and applying partial integration in the last two terms of (69), we get a vanishing contribution of $t_{3,\text{inhom.}}^n$ to $V_{(4)}$:

$$
\int d\rho \frac{\cosh^2 \rho}{\sinh \rho}(\sinh^2 \rho \, \dot{t}_{1,+}^n(\rho)\dot{t}_{3,\text{inhom.}}^{-n}(\rho) + n^2 t_{1,+}^n(\rho)t_{3,\text{inhom.}}^{-n}(\rho))
$$

$$
= |n|^2 \int_0^\infty d\rho \frac{t_{1,+}^{-n}(\rho)g_{-n}(\rho)}{\sinh \rho} \left( -\frac{1}{|n|} + |n| + \left( \frac{1}{|n|} + |n| + \cosh \rho + \frac{1}{\cosh \rho} \right)\frac{\cosh \rho - |n|}{\cosh \rho + |n|} \right.
$$

$$
\left. - \cosh \rho + \frac{1}{\cosh \rho} \right)
$$

(73)

$$
= 0.
$$

From the remaining contribution of the homogenous term in the solution of the equation of motion, we obtain in total

$$
V_{(4)} = -\int d\rho \, d\phi \left( \frac{\cosh^2 \rho}{\sinh \rho} \left[ \sinh^2 \rho \, \dot{t}_1 \dot{t}_3 + t_1' t_3' + \frac{\sinh^2 \rho \, \dot{t}_2 \dot{t}_2 + t_2' t_2'}{2} \right] \right.
$$

$$
\left. + \frac{\cosh^4 \rho}{8 \sinh^3 \rho} \left[ \sinh^2 \rho \, \dot{t}_1 \dot{t}_1 + t_1' t_1' \right]^2 \right)
$$

$$
= 2\pi \sum_n (|n|^3 - |n|) \left( C_1^n C_3^{-n} + \frac{1}{2} C_2^n C_2^{-n} \right)
$$

(74)

$$
+ \frac{\pi}{4} \sum_{n,m,r} |n||m||n+r||m-r|C_n^1 C_m^1 C_{r-m}^1 C_{-n-r}^1
$$

$$
\times \left[ \alpha_1^{n,m,r} \left( \sum_{i=1}^k \frac{(-1)^{k-i}}{i} + (-1)^k \log 2 \right) + \alpha_2^{n,m,r} \right],
$$

where we have used the shorthand notation $k = |n| + |m| + |r - m| + |-n - r|$ and

$$
\begin{aligned}
\alpha_1^{n,m,r} =& \frac{1}{3}(|m|^3 - |m| + |n|^3 - |n| + |r - m|^3 - |r - m| + |n + r|^3 - |n + r|) \\
& + 2m|n|(n - 1/n) + 2n|m|(m - 1/m),
\end{aligned}
\tag{75}
$$

$$
\begin{aligned}
12\alpha_2^{n,m,r} =& 4(1 - k^2) - 3(-1)^{k/2}(k - k^3) \\
& + (9(-1)^{k/2}k - 12)(2mn - |m - r||n + r| - (|n| + |m|)(|n + r| + |m - r|) - |m||n|) \\
& + \frac{1}{4k - k^3}\Big[-120 + 28k^2 - 4k^4 \\
& + (72 - 12k^2)(2mn - |m - r||n + r| - (|n| + |m|)(|n + r| + |m - r|) - |m||n|) \\
& + 12\frac{|m||n||m - r||n + r|}{mn(m - r)(n + r)}\Big(6 - 7k^2 + k^4 \\
& + (2 - k^2)(|n||m| + (|n| + |m|)(|m - r| + |n + r|) + |m - r||n + r|) \\
& - (4 - 2k^2)(m - r)(n + r) - k|n||m|(|m - r| + |n + r|)\Big) \\
& + (9(-1)^{k/2}(4k - 4k^3) - 12(4 - k^2) - 12k)\Big(-2m|n|/n - 2n|m|/m \\
& + (|n| + |m|)|m - r||n + r| - (2mn - |m||n|)(|m - r| + |n + r|)\Big) \\
& + 12\frac{|m||n|}{mn}\Big(2k(|m - r| + |n + r|) + k(|n| + |m|)(2|m - r||n + r| - (m - r)(n + r)) \\
& + 4 + 2(2|n||m| - mn)|m - r||n + r| + 4(|n| + |m|)(|n + r| + |m - r|) \\
& - 2|m||n|(m - r)(n + r)\Big)\Big].
\end{aligned}
\tag{76}
$$

## B  Fubini-Study complexity

To compare our gravity results for the "complexity=volume" proposal with the Fubini-Study complexity of [12,13], we now extend the calculation of the complexity in [12,13] to general perturbative conformal transformations up to fourth order in perturbation theory.

The complexity functional of [12,13] is given by

$$
C_{\text{FS}} = \int ds \left( \langle Q(s)^2 \rangle - \langle Q(s) \rangle^2 \right),
\tag{77}
$$

with the circuit generator $Q$ from (2). Let us treat the circuit (a) first. In this case $Q = Q_{(a)}$ and

$$
C_{\text{FS}} = \int ds \int \frac{dx\,dy}{4\pi^2} \Pi(x, y) \frac{\dot{f}(s, x)}{f'(s, x)} \frac{\dot{f}(s, y)}{f'(s, y)},
\tag{78}
$$

where

$$
\Pi(x, y) = \langle T(x)T(y) \rangle - \langle T(x) \rangle \langle T(y) \rangle = \frac{c}{32\sin^4((x - y)/2)} - \frac{h}{2\sin((x - y)/2)^2}.
\tag{79}
$$

The corresponding complexity is determined by minimising (77). Thus we need to solve the

equations of motion

$$
\int dx\Bigg[\Bigg(-\frac{\ddot{f}(s,x)}{f'(s,x)f'(s,y)}+\frac{\dot{f}(s,x)\dot{f}'(s,x)}{f'(s,x)^2 f'(s,y)}+2\frac{\dot{f}(s,x)\dot{f}'(s,y)}{f'(s,x)f'(s,y)^2}
$$
$$
-2\frac{\dot{f}(s,y)\dot{f}(s,x)f''(s,y)}{f'(s,x)f'(s,y)^3}\Bigg)\Pi(x,y)+\frac{\dot{f}(s,x)\dot{f}(s,y)}{f'(s,y)^2 f'(s,x)}\partial_y\Pi(x,y)\Bigg]=0\,. \tag{80}
$$

This is achieved perturbatively. We expand

$$
f(s,x)=x+\varepsilon f_1(s,x)+\varepsilon^2 f_2(s,x)+\mathcal{O}(\varepsilon^3)\,, \tag{81}
$$

and determine the solution of (80) order by order in $\varepsilon$. Without loss of generality we take $s\in[0,1]$ and impose the boundary conditions

$$
f(0,x)=0\,,\quad f(1,x)=f(x)\,, \tag{82}
$$

where the final transformation $f(1,x)$ is the conformal transformation that yields the Bañados geometry in the dual bulk picture. Note that in agreement with the gravity result, the first order contribution (in $\varepsilon$) to the complexity vanishes.

**Second order**  In this case, the solution of the equations of motion is given by a linearly increasing function in the circuit time parameter $s$,

$$
\int dx\,\ddot{f}_1(s,x)\Pi(x,y)=0\qquad\Rightarrow f_1(s,x)=s f_1(x)\,. \tag{83}
$$

Hence we obtain the following complexity[9]

$$
\mathcal{C}_{(2)}=\mathcal{C}_{\mathrm{FS}}\big|_{\mathcal{O}(\varepsilon^2)}=\int ds\int\frac{dx\,dy}{4\pi^2}\Pi(x,y)\dot{f}^1(s,x)\dot{f}^1(s,y)
$$
$$
=\sum_n\Big(\frac{c}{24}(|n|^3-|n|)+h|n|\Big)f_1^n f_1^{-n}\,, \tag{84}
$$

where $f_1^n$ is the $n$-th Fourier mode of $f_1$. For $h=0$ and $f_1^n=C_1^n$, (84) is proportional to the "complexity=volume" result (56) from the gravity theory.

**Third order**  To this order, we get a solution that is quadratic in $s$,

$$
\int dx\,\ddot{f}_2(s,x)\Pi(x,y)
$$
$$
=\int dx\Big[\dot{f}_1(s,x)(\dot{f}_1'(s,x)+2f_1'(s,y))\Pi(x,y)+\dot{f}_1(s,x)\dot{f}_1(s,y)\partial_y\Pi(x,y)\Big]
$$
$$
=\int dx\Big[f_1(x)f_1(y)\partial_y\Pi(x,y)+f_1(x)(f_1'(x)+2f_1'(y))\Pi(x,y)\Big] \tag{85}
$$
$$
\Rightarrow f_2(s,x)=\frac{1}{2}A_2(x)s^2+B_2(x)s+C_2(x)\,.
$$

---

[9]Note that to evaluate the $x$ and $y$ integrals in (77), a regularisation procedure is necessary [12,13]. Concretely, we use differential regularisation to write the $1/\sin((x-y)/2)$ terms in (79) as derivatives of $\log[\sin((x-y)/2)^2]$ and shift these derivatives onto the prefactor of the $\Pi(x,y)$ term in (77) by partial integration (see [12,13] for details).

The boundary conditions $f_2(0,x) = 0$, $f_2(1,x) = f_2(x)$ fix $C_2(x) = 0$, $\frac{1}{2}A_2(x) + B_2(x) = f_2(x)$. We then obtain for the complexity to second order

$$
\begin{aligned}
C_{(3)} &= C_{\text{FS}}|_{\mathcal{O}(\varepsilon^3)} \\
&= \int ds \int \frac{dx\,dy}{4\pi^2} \Pi(x,y)(2\dot{f}_1(s,x)\dot{f}_2(s,y) - \dot{f}_1(s,x)\dot{f}_1(s,y)(f_1'(s,x)+f_1'(s,y))) \\
&= 2\sum_n \left(\frac{c}{24}(|n|^3 - |n|) + h|n|\right) f_1^n f_2^{-n} - i\sum_{n,m} m\left(\frac{c}{24}(|n|^3 - |n|) + h|n|\right) f_1^n f_1^m f_1^{-n-m}.
\end{aligned}
\tag{86}
$$

Again, for $h = 0$ and $C_2^n = f_2^n - i\sum_m m f_1^m f_1^{n-m}$ this is proportional to the gravity result (59). Both the field theory complexity functional and the gravity result are invariant under replaing the transformation $f(x)$ by its inverse $F(x)$ (for $C_{(2)}$ and $C_{(3)}$ this amounts to replacing $f_1(x) \to -f_1(x)$ and $f_2(x) \to -f_2(x) + f_1'(x)f_1(x)$).

**Fourth order**   The equation of motion leads to a solution of $f(s,x)$ to third order with a third order polynomial in $s$,

$$
f_3(s,x) = \frac{1}{6}A_3(x)s^3 + \frac{1}{2}B_3(x)s^2 + sC_3(x) + D_3(x).
\tag{87}
$$

The boundary conditions $f_3(0,x) = 0$, $f_3(1,x) = f_3(x)$ determine enough of $f_3(s,x)$ to be able to compute $C_{(4)}$. However, for $C_{(4)}$ we also need to solve the second order e.o.m. (85). This is readily accomplished by using the Fourier decomposition of $A_2(x)$. Then (85) is equivalent to

$$
\begin{aligned}
\int &dz\,dx\,\Pi(x,z)A_2(x)e^{-inz} = (2\pi)^2\left(\frac{c}{24}(|n|^3 - |n|) + h|n|\right)A_2^n \\
&= \int dz\,dx[inf_1(x)f_1(z) + f_1(x)(f_1'(x) + f_1'(z))]\Pi(x,z)e^{-inz} \\
\Rightarrow A_2^n &= \frac{-i}{\frac{c}{24}(|n|^3 - |n|) + h|n|}\Bigg[\sum_r f_1^r f_1^{n-r}\Big(\frac{c}{24}(|r|(2n - n^3 - r + r^3 + 2nr(n-r)) \\
&\qquad\qquad\qquad + |n|(1 - n^2)(n-r)) - h(|r|(2n - r) + |n|(n-r))\Big)\Bigg].
\end{aligned}
\tag{88}
$$

The complexity is then given by

$$
C_{(4)} = C_{\text{FS}}|_{\mathcal{O}(\varepsilon^4)} = C_{(4)}^A + C_{(4)}^B + C_{(4)}^C,
\tag{89}
$$

where

$$
\begin{aligned}
C_{(4)}^C &= \int \frac{dx\,dy}{4\pi^2}\Pi(x,y)[f_1(x)f_3(y) + f_3(x)f_1(y)] \\
&= 2\sum_n \left(\frac{c}{24}(|n|^3 - |n|) + h|n|\right) f_1^n f_3^{-n},
\end{aligned}
\tag{90}
$$

$$
\begin{aligned}
C_{(4)}^B &= \int \frac{dx\,dy}{4\pi^2}\Pi(x,y)\Bigg[f_2(x)f_2(y) - \frac{1}{2}f_1(x)f_1(y)(f_2'(x) + f_2'(y)) \\
&\qquad\qquad - \frac{1}{2}(f_1(x)f_2(y) + f_2(x)f_1(y))(f_1'(x) + f_1'(y))\Bigg] \\
&= \sum_n \left(\frac{c}{24}(|n|^3 - |n|) + h|n|\right)\Big[f_2^n f_2^{-n} - \sum_m im(f_1^n f_2^m f_1^{-n-m} + f_1^n f_1^m f_2^{-n-m} \\
&\qquad\qquad\qquad\qquad + f_2^n f_1^m f_1^{-n-m})\Big],
\end{aligned}
\tag{91}
$$

and

$$
\begin{aligned}
C^A_{(4)} &= \int \frac{dx\,dy}{4\pi^2} \Pi(x,y) \Big[ \frac{1}{12} A_2(x) A_2(y) - \frac{1}{12}(A_2(x) f_1(y) + f_1(x) A_2(y))(f_1'(x) + f_1'(y)) \\
&\qquad\qquad + \frac{1}{12} f_1(x) f_1(y)(A_2'(x) + A_2'(y)) + \frac{1}{3} f_1(x) f_1(y)(f_1'(x) + f_1'(y))^2 \Big] \\
&= \sum_{m,n,r} \frac{c}{24} f_1^m f_1^n f_1^{m-r} f_1^{-n-r} \Big[ \\
&\quad -\frac{1}{12}\frac{1}{|r|^3 - |r|}\Big(|n|(n^3 - n + r^3 + 2nr(n+r) - 2r) + (n+r)|r|(r^2 - 1)\Big) \\
&\qquad\qquad \Big(|m|(m^3 - m - r^3 + 2mr(r-m) + 2r) + (m-r)|r|(r^2 - 1)\Big) \\
&\quad -\frac{1}{12}\Big((m+2n)|m|(m^2 - 1) + (n+2m)|n|(n^2 - 1) + (m+n)|m+n|((m+n)^2 - 1)\Big) \\
&\qquad \frac{1}{|m+n|^3 - |m+n|}\Big( (n+r)|m+n|((m+n)^2 - 1) \\
&\qquad\qquad + |m-r|\big(r((n+r)^2 - 1) + n((m+n)^2 - 1) \\
&\qquad\qquad\qquad - (m+n) + (n^2 - mr)(r-m)\big)\Big) \\
&\quad +\frac{1}{3}(m-r)(n+r)(|m|(m^2 - 1) + |n|(n^2 - 1) + |r|(r^2 - 1))\Big] + \text{terms proportional to } h.
\end{aligned}
\tag{92}
$$

Comparison with (74) clearly shows that the field theory complexity functional (77) does not match with the "complexity=volume" result from the gravity theory for general conformal transformations to fourth order in perturbation theory.

To derive the Fubini-Study complexity for the circuit (b), we simply replace $Q_{(a)}$ by $Q_{(b)}$. An analogous calculation to the one above for the circuit (a) shows that although this replacement changes the optimal path in the circuit as expected, the value of the Fubini-Study complexity functional is unchanged up to the third order.

Finally, let us note that the Fubini-Study complexity functional (77) is not unique in the sense that any complexity functional defined as a time-integral of a function of the Fubini-Study metric has the same optimal path as (77),

$$
C_{\text{FS,generalized}} = \int ds\, \alpha\left(\sqrt{\langle Q(s)^2\rangle - \langle Q(s)\rangle^2}\right).
\tag{93}
$$

Here $\alpha$ is a positive function. Equation (77) is therefore only one particular member of a more general family obtained by choosing $\alpha(x) = x^2$. Our analysis can also exclude that other member of this family match with the volume change in the "complexity=volume" prescription. To see this, expand the function $\alpha(x)$ in a power series in $x$. The only term in this expansion that gives an $\mathcal{O}(\varepsilon^4)$ contribution to the complexity but no $\mathcal{O}(\varepsilon^3)$ and $\mathcal{O}(\varepsilon^2)$ contributions is the $x^4$ term. However by explicit calculation it is easy to see that this term together with the $\mathcal{O}(\varepsilon^4)$ contributions from $x^2$ or $x^3$ terms cannot give a result equal to the fourth order term (74) in the perturbation series in $\varepsilon$ of the volume change.

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
