# Peer review of "Exact Gravity Duals for Simple Quantum Circuits"

_SciPost Physics, doi:SciPost Phys. 13, 061 (2022)_

## Round 1 · Referee Report · Anonymous (Referee 1) · 2022-5-19

Strengths

addresses important and timely question(s)
distinguishes abstract trajectories in Hilbert space from circuits in physical time
explains how to translate a CFT cost function into the bulk
disproves a prior conjecture / guess
clearly written
well organized
easy to understand

Weaknesses

no weaknesses

Report

The paper definitely meets all criteria and I recommend publication.

The manuscript is about holographic complexity. It studies the Fubini-Study metric cost (technically, cost = FS metric^2) for circuits in the two-dimensional conformal group. As a first big novelty, the paper distinguishes abstract circuits (as trajectories in Hilbert space) from physical circuits (which progress in physical time). This is a very interesting and important development.

The paper calculates the complexity in both types of circuits and shows that it is different from bulk volume, albeit only at four order in perturbation theory! This is a very important data point for trying to match bulk notions of complexity with boundary definitions. The paper also sketches how one might try to derive a boundary cost function to match a given bulk notion of complexity.

Requested changes

(1) typo "iin" above equation (40)
(2) perhaps it would be useful to explain the factor of (-i) in the last term in equation (40). This is a difference of volumes. Is it real? If so, what cancels the (-i)?

---

## Round 1 · Referee Report · Anonymous (Referee 2) · 2022-5-20

Strengths

  • The proof that the Fubini-Study-squared functional cannot account for the volume differences, as suggested in a previous paper

Weaknesses

  • Unclear to what extent the paper contributes to our knowledge on the subject of complexity in CFT/holography

Report

Within the framework of two-dimensional CFTs, the authors consider “circuits” of states obtained by acting on the vacuum with unitary operators $U(\tau)$ constructed from the Virasoro algebra generators. The circuits are parametrised by a real-valued parameter $\tau$, which parametrises a path in the space of unitary operators. Infinitesimal motion along the path is produced by an operator $Q(\tau)$ (sometimes called “instantaneous Hamiltonian” in the literature) which is chosen to be proportional to the stress-tensor. Then, for each value of $\tau$, the circuit implements a conformal transformation. Different choices give rise to different circuits. In the paper, the authors consider two circuits. For the first, the evolution occurs for states which live on the same time slice of different spacetimes, whereas for the second, the circuit parameter is chosen to be the physical time and the circuit produces states which live on different slices of the same spacetime.

The authors propose to identify the instantaneous Hamiltonian of both circuits with the physical Hamiltonian of the CFT living in some background metric. For each circuit, the authors compute the background metric and the stress tensor expectation values, finding that both circuits are defined in flat space and that the relevant transformations correspond to different time foliations (different depending on the circuit).

Then, the authors propose to reconstruct an holographic bulk metric for each circuit by using the usual Fefferman-Graham expansion and plugging the resulting values of the boundary metric and stress-tensor expectation values. From this perspective, each circuit would correspond to a particular time-slicing of pure AdS$_3$.

In a different section, the authors show that a notion of complexity based on a cost function given by the square of the Fubini-Study cost (a notion of complexity often used in the literature) applied to the circuits does not generally equal the change in the volume of the extremal slice (with respect to the vacuum state one), disproving a previous conjecture.

I think the paper does contain some novel results, although I am not totally sure to what extent it contributes to the bulk of knowledge already present in the literature in the area of CFT complexity and/or holography. Some of the proposals which are central to the paper, like the construction of an auxiliary bulk metric for the Virasoro protocols of a given 2d CFT by imposing the physical Hamiltonian to be the instantaneous Hamiltonian, seem more or less straightforward consequences of the holographic dictionary (applied in this case in a slightly modified framework). On the other hand, while I think it is interesting to know that the connection between the Fubini-Study-squared functional does not reproduce the volume differences, I should stress that such a notion does not really correspond to a complexity functional in the Nielsen sense, due to the square in the cost function, which spoils its scaling in various situations. Hence, the scope of this result is a bit unclear.

In sum, I remain agnostic on whether or not the paper should be accepted for publication in SciPost.
  • validity: ok
  • significance: ok
  • originality: ok
  • clarity: ok
  • formatting: good
  • grammar: excellent

Author:  Marius Gerbershagen  on 2022-05-25  [id 2527]

(in reply to Report 2 on 2022-05-20)

We believe that our paper significantly contributes to the research on computational complexity measures in AdS/CFT, in particular through the following results:

Most importantly, we provide a gravitational realization of an important class of quantum circuits built out of conformal transformations that features prominently in various studies of computational complexity in conformal field theories (see e.g. arxiv:1807.04422, arxiv:2103.06920, arxiv:2004.03619, arxiv:2005.02415, arxiv:1806.08376).

While this construction is indeed an application of the holographic dictionary, it is still a non-trivial new approach in the following respects: As the other report notes, in section 3 we carefully distinguish constructions that treat the circuit parameter $\tau$ as a physical time - both in the field theory and in the gravity dual - from those that treat it as an auxiliary parameter, present in addition to physical time. We provide gravity dual realizations for both types of circuits in section 3.2 and 3.3. Moreover, we point out that it is sufficient to use flat boundary metrics and explain why contact terms in correlation functions do not play a role in our construction (see page 6). To our knowledge, these extensions and clarifications, as well as constructions of gravity duals to quantum circuits in general, have not appeared before in the literature.

More generally, our construction provides a starting point for further investigations as it provides the basis for mapping cost functions for field theory complexity measures to bulk quantities and vice-versa. In this regard, the fact that our construction is an application of the holographic dictionary is an advantage, since this allows for performing this mapping from first principles and actually deriving gravity duals to computational complexity measures using the AdS/CFT dictionary.

We made a first step in this direction for the (squared) Fubini-Study cost function. This cost function does not fit into the original computational complexity framework proposed by Nielsen, as the referee correctly points out. However, we believe that only studying the framework proposed by Nielsen is too restrictive and that there is value in also considering other cost functions in the AdS/CFT context. Such cost functions have also appeared in many previous studies of computational complexity in QFT.

An example for this is the squared Fubini-Study distance. Even though it does not fit in the original Nielsen framework, it has nevertheless shown to be related to the complexity=volume proposal at leading order in perturbation theory. The result that these connections do not extend to higher order in perturbation theory provides a further significant contribution to the computational complexity research direction in holography. This result is logically distinct from previous results that the squared Fubini-Study cost function does not yield a complexity measure in the sense of Nielsen since the complexity=volume proposal does not presuppose a Nielsen-type complexity.

In summary, our paper presents a new approach to the study of computational complexity in AdS/CFT based on using the AdS/CFT dictionary to map complexity measures or cost functions to bulk quantities from first principles. This approach is implemented in a concrete setting and applied to investigate the gravity dual to the Fubini-Study cost function that is well-studied in field theory.

---

## Round 2 · List of Changes

• implemented the requested changes around equation (40)
  • fixed some typos in appendix A

---

## Editorial Decision

published